## THE NATURAL HISTORY OF MODEL ORGANISMS

# The biology of *C. richardii* as a tool to understand plant evolution

**Abstract:** The fern *Ceratopteris richardii* has been studied as a model organism for over 50 years because it is easy to grow and has a short life cycle. In particular, as the first homosporous vascular plant for which genomic resources were developed, *C. richardii* has been an important system for studying plant evolution. However, we know relatively little about the natural history of *C. richardii*. In this article, we summarize what is known about this aspect of *C. richardii*, and discuss how learning more about its natural history could greatly increase our understanding of the evolution of land plants.

**SYLVIA P KINOSIAN\* AND PAUL G WOLF**

**\*For correspondence:**
sylvia.kinosian@gmail.com

**Competing interest:** The authors declare that no competing interests exist.

## Introduction

The genus *Ceratopteris* has a long and complicated taxonomic history. It was first described by Linnaeus under the genus *Acrostichum* (*Linnaeus, 1764*), and the name *Ceratopteris* was later assigned by Brongniart (*Brongniart, 1821*). Since then, *Ceratopteris* has been placed in a number of different families, with the number of species within the genus ranging between one and twelve (*Lloyd, 1974*). Today it is placed within Pteridaceae, one of the largest and most diverse fern families (*PPG, 2016*; *Figure 1*).

There are about ten species within *Ceratopteris*, which can be found throughout the tropics (*Figure 2*; *Masuyama and Watano, 2010*; *Zhang et al., 2020*; *Yu et al., 2021*). The classification of these species was made difficult by their inconsistent morphologies, and molecular methods were needed to reconstruct a backbone phylogeny for the genus (*Adjie et al., 2007*; *Kinosian et al., 2020a*). Recent work has shown that cryptic and hybrid species may be quite common in *Ceratopteris*, warranting a more rigorous evaluation of the relationships between species in the genus (e.g., *Kinosian et al., 2020b*).

*Ceratopteris richardii* was first developed as a model system for ferns in the 1960s and 70s, primarily because it was easy to grow in the lab and had a short life cycle (*Figure 2*; *Pal and Pal, 1962*; *Pal and Pal, 1963*; *Klekowski,*

*1970*; *Stein, 1971*; *Hickok, 1973*; *Hickok and Klekowski, 1973*; *Lloyd and Warne, 1978*). Many studies used spores from a Cuban vouchered collection, now known as the Hnn strain or C-fern (*Hickok, 1977*). Additional strains of *C. richardii* and the species *C. thalictroides* and *C. pteridoides* have since been developed (*Hickok and Klekowski, 1974*; *Nakazato et al., 2006*; *Muthukumar et al., 2013*). In the past few decades, *Ceratopteris* has become an important model in the study of sex determination (*Eberle et al., 1995*; *Ganger et al., 2019*; *Atallah et al., 2018*; *Banks, 1997*), apogamy (*Bui et al., 2017*; *Cordle et al., 2010*), genome structure (*Nakazato et al., 2006*; *Baniaga and Barker, 2019*), hybridization (*Hickok and Klekowski, 1974*; *Adjie et al., 2007*), reproductive barriers (*Nakazato et al., 2007*), developmental biology (*Hou and Hill, 2002*; *Conway and Di Stilio, 2020*; *Sun and Li, 2020*; *Aragón-Raygoza et al., 2020*), and transgenic studies in ferns (*Plackett et al., 2018*; *Bui et al., 2015*; *Cannon et al., 2018*).

*Ceratopteris richardii* is one of few spore-bearing model systems (e.g., *Physcomitrella*: *Cove, 2005*; *Rensing et al., 2008*; *Selaginella*: *Banks et al., 2011*) and the only vascular spore-bearing homosporous model species, making it a critical evolutionary lineage in comparative studies (see *Box 1* for a glossary of terms used in this article). Compared to other model plants,

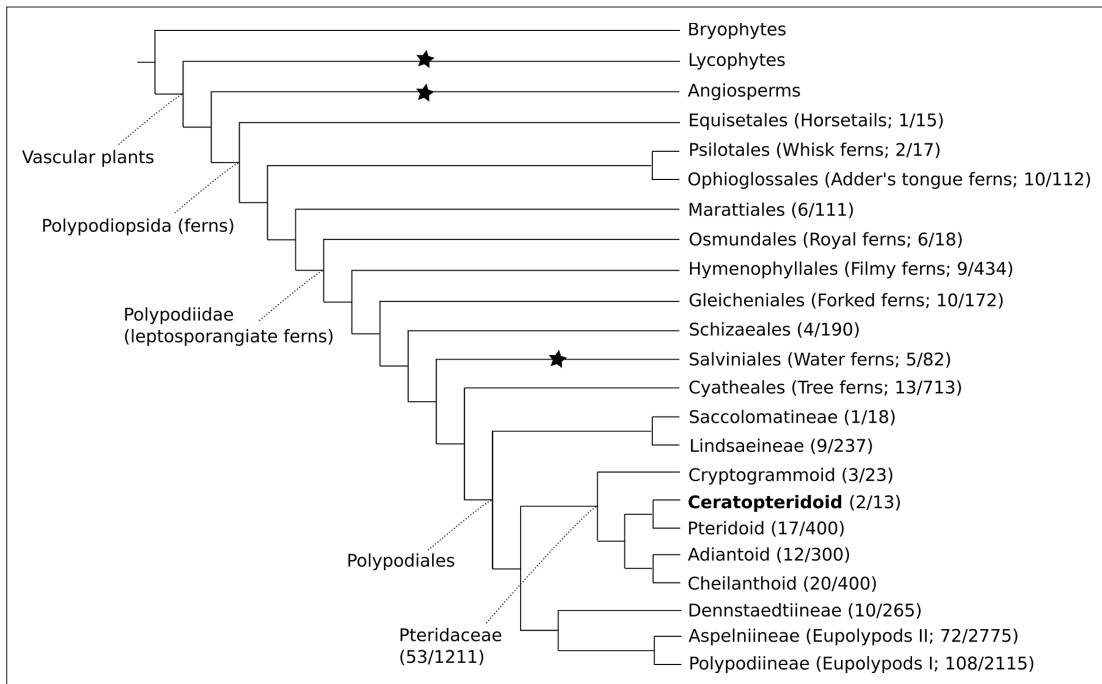

**Figure 1.** Cartoon phylogeny of land plants. Within the Polypodiopsida (ferns), the estimated number of genera and species (genera/species) are noted for each major group. Black stars show the three extant independent evolutions of heterospory (***Bateman and DiMichele, 1994***). In the family Pteridaceae, the five major groups are shown, including the Ceratopteridiode clade (in bold) which includes the genera *Acrostichum* and *Ceratopteris*. Within *Ceratopteris*, there are about ten species found throughout the world's tropics (see ***Figures 2 and 3***).

*Ceratopteris* has a large genome (~11 GB) and high base chromosome number (n = 39), which has partly caused genetic resources for *C. richardii* to lag behind those of other plant model systems, delaying such comparative work.

Recently, however, the first draft genome sequence for *C. richardii* was published (***Marchant, 2019a***), which was the first for a homosporous fern. *Ceratopteris richardii* was chosen for sequencing because, although it has a large genome compared to other plants, it is relatively small for a homosporous fern (***Sessa et al., 2014***; ***Marchant et al., 2019b***).

Having a reference genome for *C. richardii* expands its research potential and builds on decades of previous work. A homosporous plant genome provides the opportunity for exploring and comparing various aspects of plant biology such as the alternation of generations, sex determination, and reproductive modes between heterosporous and homosporous plants. In addition, a reference genome for *Ceratopteris* is beneficial for the development of new markers in targeted gene sequencing or whole-genome resequencing. In turn, this makes incorporating wild collections into genomic research much easier and will help us gain a more nuanced understanding of the biology, ecology, and evolutionary history of *Ceratopteris*.

## The variable natural history of *Ceratopteris*

The model species *Ceratopteris richardii* originates in the Caribbean and Western Africa, and grows rooted or floating in shallow water (***Figure 2***). Indeed, all species within *Ceratopteris* grow in or near areas in the tropics that become inundated seasonally (***Figures 2 and 3***), mostly growing in fresh water, though they can tolerate salt water (***Lloyd, 1974***; ***Warne and Hickok, 1987***).

Its sister genus, *Acrostichum*, is well-known for being able to tolerate high levels of salt as it grows in tidal and intertidal habitats (***Zhang et al., 2013***; ***Medina et al., 1990***). The extent of natural salt tolerance in *Ceratopteris* is not fully understood, but salt-tolerant mutants of *C. richardii* are easy to generate in the lab (***Chasan, 1992***; ***Warne et al., 1995***). Continuing to study salt tolerance in *Ceratopteris* may be beneficial in understanding the genetic mechanisms of this trait, or applying such findings to crop systems in the future.

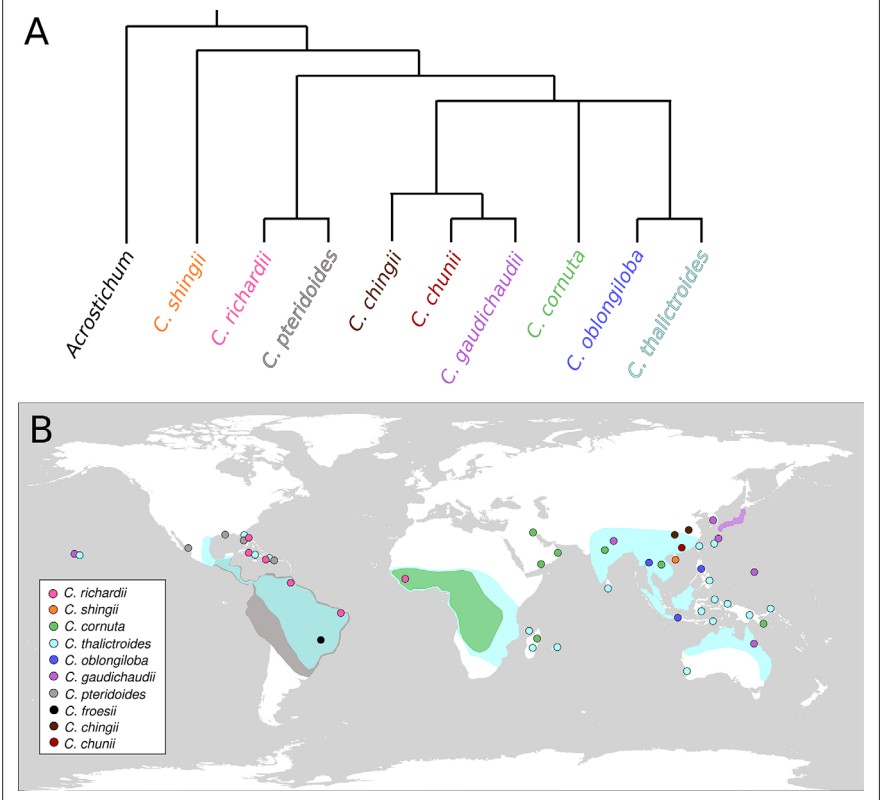

**Figure 2.** Phylogenetic and geographic representation of the genus *Ceratopteris*. (**A**) Phylogenetic reconstruction of *Ceratopteris*, with the sister genus *Acrostichum* as the outgroup, based on **Kinosian et al., 2020a**; **Adjie et al., 2007**; **Yu et al., 2021 Zhang et al., 2020**. Absent from this phylogeny is the Brazilian species *C. froesii*, for which no genetic sequence data is available. (**B**) Distribution map of ten *Ceratopteris* species: pink, *C. richardii*; orange, *C. shingii*; green, *C. cornuta*; light blue, *C. thalictroides*; dark blue, *C. oblongiloba*; purple, *C. gaudichaudii*; grey, *C. pteridoides*; black, *C. froesii*; brown, *C. chingii*, and red *C. chunii*. Shaded areas show where a species is common, colored dots show where species occur outside their most common range, or multiple species are found in a small area. Location data from **Kinosian et al., 2020a**; **Lloyd, 1974**; **Masuyama and Watano, 2010**; **Zhang et al., 2020**; **Yu et al., 2021** https://gbif.org/.

Through much of its range, *Ceratopteris* inhabits ephemeral water sources. To reproduce in this fleeting habitat *C. richardii* has a short life cycle of about 120 days (**Stein, 1971**), which is much shorter than almost all ferns' annual or multi-year life cycles. The fern life cycle, like that of all land plants, is characterized by the alternation of generations between the diploid sporophyte and haploid gametophyte. Following fertilization, a diploid zygote is formed that is temporarily reliant on the gametophyte and becomes self-sufficient over time. In the fern life cycle, the gametophyte and sporophyte generations are completely separate at certain stages. Comparatively, in bryophytes, the sporophyte generation is entirely dependent on the gametophyte, and the opposite is true in seed plants. The fern life cycle provides an opportunity to study sporophyte and gametophyte generation separately, something that is not possible in other lineages of plants.

Because ferns have independent and free-living sporophytic and gametophytic phases, there are multiple ways in which the life cycle of *Ceratopteris* can proceed (**Haufler et al., 2016**). Fertilization can occur via gametes from different plants (sporophytic outcrossing), gametes from the same plant but different gametophytes (gametophytic outcrossing), and also gametes from the same gametophyte (gametophytic selfing). Studies show evidence of outcrossing within (**Nakazato et al., 2007**) and among species of *Ceratopteris* (i.e., hybridization, **Adjie et al., 2007**; **Kinosian et al., 2020a**; **Hickok and Klekowski, 1974**).

Outcrossing and gametophytic sex expression in ferns is often controlled by pheromones known as antheridiogens. These gibberellin-related chemicals are released by early-germinating archegoniate gametophytes to promote antheridiate gametophyte development in immature gametophytes. This pheromone system confers some of the benefits of heterospory to homosporous plants, namely outcrossing (**Bateman and DiMichele, 1994**; **Hornych et al., 2021**). In the case of *Ceratopteris*, however, solitary gametophytes can also become bisexual and self-fertilize; therefore, theoretically, only one spore can colonize new habitats (**Schedlbauer and Klekowski, 1972**). This is the case for many ferns and aids *Ceratopteris* as a model organism because studies can be designed around this flexible life history.

In addition, asexual reproduction (apomixis) can be induced in *C. richardii* in the lab (**Cordle et al., 2007**). The variation within the life cycle of *Ceratopteris* makes it a powerful system in which to study reproduction in ferns, as well as an evolutionary point of reference for understanding reproduction in seed plants.

*Ceratopteris* was developed as a model organism for many of the same reasons as other model systems. It is easy to grow in a lab setting, has a rapid life cycle that makes experiments tangible, and is tractable for genetic transformations (**Eberle et al., 1995**; **Hickok et al., 1995**). Model organisms are often chosen for convenience, but that can make them poor representative taxa (**Alfred and Baldwin, 2015**).

In the case of *Ceratopteris*, it has several traits that are very unusual among ferns. For example, it is one of only a handful of semi-aquatic species out of around 12,000 extant ferns (**PPG, 2016**). The life cycle of *Ceratopteris*, while beneficial

for lab experiments, is incredibly short for a fern (*Stein, 1971*). Finally, it has half or a quarter of the number of spores typical for a leptosporangiate fern: most leptosporangiate ferns produce 64 spores per sporangium, whereas species in *Ceratopteris* produce 32 or 16 spores per sporangium (*Lloyd, 1974*). This is important because a spore number of 32 or 16 is often indicative of apogamy (*Grusz, 2016*), but no natural apogamous taxa have been described in *Ceratopteris*. These characteristics, among others, make *Ceratopteris* a good model species but not necessarily an accurate representation of all ferns.

## A transgenic model for seed-free plants

Free-living generations and a flexible life cycle make *Ceratopteris* an important model for evolutionary developmental studies. A reference ontogenetic framework for the Hnn strain of *C. richardii* (C-fern; *Hickok et al., 1995*) was recently published, detailing the development of the gametophyte and sporophyte, providing an important reference for future work (*Conway and Di Stilio, 2020*). This reference, in combination with stable transformation techniques, plus a *C. richardii* transcriptome (*Geng et al., 2021*; *Atallah et al., 2018*) and genome (*Marchant et al., 2022*; *Marchant et al., 2019b*) now provide the necessary suite of tools for comparative work.

In the Hnn strain of *Ceratopteris richardii* stable transgenic lines have been established in both the gametophyte and sporophyte generations. Transformation of the tissue in these plants has been accomplished by bombardment of sporophytic tissue by tungsten microparticles (*Plackett et al., 2014*; *Plackett et al., 2015a*), *Agrobacterium* infection of haploid gametophyte tissue (*Bui et al., 2015*), and agrobacterium infection of haploid spores (*Muthukumar et al., 2013*).

Transformation on gametophytes provides an important perspective on gene function in the haploid generation, something that is not as easy in seed plants. Another benefit of working with transformed gametophytes is that they can self-fertilize to produce sporophytes that are stable homozygotes. However, this can also be accomplished if transformation is done on sporophytes. Spores can be collected and screening of resulting gametophytes can then be used to produce stable homozygous transgenic lines via gametophytic selfing.

Thus, transgenic lines of *C. richardii* have been developed, and allow for comparative studies of gene function and evolution across land plants in both the gametophyte and sporophyte generations. The ability to have transgenic lines in both generations provides a unique perspective for studying how genes, growth conditions, or other factors affect sporophytes and gametophytes differently.

A recent study using both transgenic gametophytes and sporophytes of *Ceratopteris* investigated the role of the LEAFY transcription factor (LFY) in development (*Plackett et al., 2018*). LFY is important for cell division in moss embryos (*Tanahashi et al., 2005*) and angiosperm floral meristem development (*Carpenter and Coen, 1990*). While it is known to be important in both of these lineages, there is no functional overlap between mosses and angiosperms, so understanding the evolutionary history of LFY has been challenging.

*Ceratopteris* provides an evolutionary and functional midpoint with which to study the role of the LFY gene in development. Using several transgenic lines of *C. richardii,* Plackett et al. evaluated the role of LFY in sporophyte development. They also reported, for the first time in any land plant, that LFY function is required for *C. richardii* gametophyte development (*Plackett et al., 2018*). This suggests that LFY was important for gametophyte development for the last common ancestor of ferns and seed plants, but this function was lost in seed plants where the gametophyte is greatly reduced (*Plackett et al., 2018*). Incorporating other vegetative developmental systems into further studies with *C. richardii* (e.g., *Vasco et al., 2013*; *Vasco et al., 2016*; *Hernández-Hernández et al., 2021*) will help us gain a more nuanced understanding of these processes across land plants.

Similarly, there is a growing body of work on the developmental patterns associated with reproduction using *Ceratopteris*. It is well-established that apogamy can be induced in *C. richardii* (*Cordle et al., 2007*), but the genetic mechanism responsible was unknown until recently. In flowering plants, BABY-BOOM (BBM) genes promote somatic embryogenesis (*Boutilier et al., 2002*; *Soriano et al., 2013*). These genes are absent in non-seed plants, but an ortholog of the BBM gene AINTEGUMENTA was identified in *C. richardii* (*Bui et al., 2017*). Using *Agrobacterium*-mediated transformations, Bui et al., created *C. richardii* gametophytes with both over- and knockdown-expression of the apogamy-inducing gene, CrANT. This was the first such study conducted in a non-seed plant and provides evidence for conserved gene

## Box 1. Glossary

Homosporous: Plants that produce one type of spore, which germinates into a gametophyte capable of producing both eggs and sperm. This group comprises most ferns, and some lycophytes, and all non-vascular plants.

Heterosporous: Plants that produce separate spores, which produce sperm and eggs respectively. This includes all seed plants, as well as a few lineages of ferns and lycophytes.

Apomixis: A form of asexual reproduction in plants. It can proceed by 'apogamy', where unreduced spores germinate into gametophytes from which sporophytic tissue can grow without fertilization; the alternative is 'apospory', where no spore is produced and a gametophyte grows directly from the parent sporophytic tissue.

Sporophyte: The diploid generation of plants that produce spores. In mosses, this generation is dependent on the gametophyte. In ferns, lycophytes, and seed plants this generation is independent.

Gametophyte: The haploid generation in plants that produce gametes. They can be 'archegoniate' (having archegonia that produce eggs) or 'antheridiate' (having antheridia that produce sperm), or be 'hermaphroditic' (producing both egg and sperm). Gametophytes are free-living in mosses, ferns, and lycophytes, but dependent on the sporophyte generation in seed plants.

Sporangium: The structure in plants that create a spore. In ferns, these are found on the underside of a leaf, often grouped in small clusters called sori.

Leptosporangiate fern: One of the major lineages of ferns, in the subclass Polypodiidae. These ferns have sporangia with long stalks that produce (typically) 64 spores, all derived from a single initial cell (**PPG, 2016**).

function in apomixis across land plant lineages (**Bui et al., 2017**).

Future work using transgenic lines of *Ceratopteris richardii* has the potential to connect gene expression and function across land plants. Work has already been done on many gene families in model bryophytes, the lycophyte *Selaginella*, and seed plants. As mentioned above, however, there is not always functional overlap between these lineages. Including *Ceratopteris* may provide such a functional or developmental link. It is important to note that ferns like *Ceratopteris* are an independent lineage with unique development characteristics that have continued to evolve since diverging from other land plants, as are bryophytes and seed plants (**Plackett et al., 2015b**; **McDaniel, 2021**). However, *Ceratopteris* does share many characteristics with bryophytes (e.g., spores, independent gametophyte generation) and seed plants (e.g., vasculature, independent sporophyte generation), which make it a key lineage to include in comparative work.

Many model and non-model plants have recently established CRISPR/Cas9 gene-editing systems. Developing CRISPR in *Ceratopteris* is promising because *C. richardii* (and the Hnn strain) is diploid with a short life cycle and homozygotes can be easily created in one generation of gametophytic selfing (**Shan et al., 2020**). One potential avenue of study with CRISPR could be the apogamy pathway in *C. richardii* (**Bui et al., 2017**), investigating the connection between apogamy and spore number in *C. richardii*. This species produces 16 spores per sporangium, a number often indicative of apogamy in ferns (**Grusz, 2016**); *C. richardii* reproduces sexually but apogamy can be easily induced. If a CRISPR system could be established in *Ceratopteris*, one might be able to extend such technology to other members of the Pteridaceae known for apogamy (**Grusz et al., 2021**; **Grusz, 2016**; **Grusz et al., 2009**), or for application in other non-model ferns.

## The environmental influence on development

*Ceratopteris richardii* is a well-known model system for studying sex determination (**Banks, 1997**) and the alternation of generations in homosporous plants (**Eberle et al., 1995**). However, there are some steps in its life cycle that are poorly understood. Almost nothing is known about how *Ceratopteris* gametophytes are established in the wild. It is unknown if

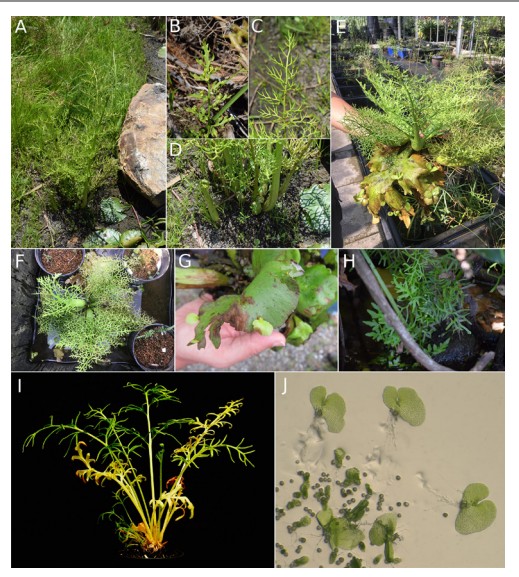

**Figure 3.** Morphological diversity of four *Ceratopteris* species. All photos by SPK unless credited otherwise. (**A**) *Ceratopteris thalictroides* mature plant from Townsville, Australia. (**B**) Detail of a sterile *C. thalictroides* leaf with buds. (**C**) Detail of a fertile *C. thalictroides* leaf. (**D**) Leaf bases and fiddleheads of *C. thalictroides*. (**E, F**) *Ceratopteris pteridoides* in cultivation at the Dr. Cecilia Koo Botanic Conservation Center in Taiwan. (**G**) Vegetative buds on *C. pteridoides* (photo by Christopher Haufler). (**H**) *Ceratopteris gaudichaudii* in Cairns, Australia. (**I**) *Ceratopteris richardii* in cultivation (photo by David Randall). (**J**) *Ceratopteris richardii* gametophytes (photo by Jo Ann Banks).

*Ceratopteris* spores must germinate on soil, or if they can germinate and establish gametophytes in standing or slow-moving water.

Recently the effect of soil bacteria on sex determination in *C. richardii* was investigated for the first time (*Ganger et al., 2019*). In the presence of a soil bacterium, there were more hermaphroditic (compared to antheridiate) gametophytes as well as increased growth (*Ganger et al., 2019*). *Ceratopteris* uses an antheridiogen pheromone system to control sex determination (*Scott and Hickok, 1987*; *Banks, 1997*), and soil bacteria may be influencing sex determination in a similar way. Additional experiments would benefit our understanding of how natural conditions might affect gametophyte establishment and sex determination in *Ceratopteris*, outside of the known role of antheridiogens. The establishment of new plants is particularly important as climate change is a threat to the current habitat of *Ceratopteris*, both as sea levels rise and rainfall becomes less predictable.

In addition to the establishment of new plants, climate change may influence the morphology, ecology, and physiology of *Ceratopteris*. There is dramatic variation in frond morphology within Japanese *C. thalictroides* based on the growing season length (*Masuyama, 1992*); such intra-species variety has not been systematically characterized in any other species in the genus. This is important to understand because leaves have been used by some authors as the primary method of identification in *Ceratopteris* (*Benedict, 1909*; *Lloyd, 1974*), despite this being one of the most variable traits. Understanding the model species *C. richardii* and its relatives in the wild is important for conservation efforts, as well as to understand what natural variation exists in these species that may be informative to future work.

## Systematics and hybridization

Hybridization among *Ceratopteris* species is well-documented (*Hickok and Klekowski, 1974*; *Nakazato et al., 2007*; *Hickok, 1977*; *Hickok, 1973*), and these hybrid taxa as well as progenitor species can be morphologically cryptic (*Adjie et al., 2007*). Lloyd predicted the presence of multiple cryptic lineages in *C. thalictroides*, but detecting these taxa was not possible at the time without genetic analysis (*Lloyd, 1974*).

During the 1990s and early 2000 s, Masuyama and colleagues examined cryptic variation within *C. thalictroides* from Asia and Oceania. They used a combination of work on allozymes and cross-breeding experiments (*Masuyama et al., 2002*), chromosome counts (*Masuyama and Watano, 2005*), morphology of wild and cultivated plants (*Masuyama and Adjie, 2008*; *Masuyama, 1992*), along with plastid and nuclear markers (*Adjie et al., 2007*) to describe three cryptic species (*Masuyama and Watano, 2010*). More recently, Zhang et al. described another cryptic species of *C. thalictroides* endemic to Hainan Province in China. This taxon, *Ceratopteris shingii*, has some unique characteristics in the genus: a creeping rhizome, terrestrial growth on volcanic rock, and is sister to all other species in the genus (*Zhang et al., 2020*). Its phylogenetic placement and unique characteristics could provide some new hypotheses for trait evolution and an updated perspective on the life history and ecology of the genus.

In addition to the diversity of *Ceratopteris* in Asia, the Americas may have novel cryptic species. Natural hybrids between *C. thalictroides* and *C. pteridoides* have been described in South

> ## Box 2. Outstanding questions about the natural history of *Ceratopteris*
>
> - What is the evolutionary history of *C. richardii*? **Kinosian et al., 2020b** were unable to find a consistent genetic identity for *C. richardii*; is this due to poor sampling, extirpation of *C. richardii* from its native range, and/or misidentification of specimens?
> - Why do some species of *Ceratopteris* produce 32 spores per sporangium, and *C. richardii* produces only 16? This is substantially less than the typical leptosporangiate fern which produces 64 spores per sporangium.
> - What is the genetic population structure of *Ceratopteris* species? Plants are typically locally abundant but regionally rare; is this due to environmental conditions, spore dispersal, or other factors? How does it affect genetic diversity across a landscape?
> - How are *Ceratopteris* gametophytes established in the wild?
> - What is the biogeographic history of the genus? How might that be influencing current species distributions and hybridization?
> - How does the habit (aquatic vs. semi-aquatic) of different *Ceratopteris* species influence population structure, breeding system, or genetic structure and function?

America (**Hickok and Klekowski, 1974**), as well as synthesized hybrids between several New World species (**Hickok, 1973**; **Hickok, 1978**). Kinosian et al., found several hybrid individuals and potentially a cryptic species of *C. thalictroides* in the Americas (**Kinosian et al., 2020a**). Interestingly, the same study did not find distinct wild populations of *C. richardii*. Future work on systematics in the group should focus on detangling cryptic species and identifying the extant range and phylogenetic placement of *C. richardii*.

A robust evolutionary tree is particularly important for *Ceratopteris* following the publication of the *C. richardii* genome. The taxonomy of model organisms is not always fully understood until after they become model systems (e.g., *Arabidopsis*, **Al-Shehbaz and O'Kane, 2002**; *Rattus norvegicus*, **Musser et al., 2005** and *Caenorhabditis elegans*, **Denver et al., 2003**; **De Ley, 2006**), and *C. richardii* is no exception.

Despite having unique characteristics like a distinct deltoid leaf shape and only 16 spores per sporangium (**Lloyd, 1974**), *C. richardii* is not often identified correctly. For example, specimens identified as *C. richardii* from Central and South America, as well as western Africa, are each more genomically similar to other species than they are to one another (**Kinosian et al., 2020b**). This could be due to misidentification of collections, a poor understanding of its native range, or the extinction of *C. richardii* in the wild. This last possible explanation is troubling because, as we discuss above, it is important to have wild populations to best understand the potential of model organisms. Revisiting the localities of known *C. richardii* collections (detailed in **Lloyd, 1974**) should be a goal for future fieldwork. New wild collections will help elucidate the outstanding questions about the taxonomy and natural history of *C. richardii*, but may also provide novel populations or strains to include in lab studies.

## Plant genome structure and evolution

On average, heterosporous plants have fewer chromosomes and smaller genomes than homosporous plants. *Ceratopteris richardii* is the first homosporous fern with the genomic resources to address why these differences between hetero- and homosporous genomes exist. Nakazato et al. generated a genetic linkage map for *C. richardii* which showed that it is likely not repeated rounds of polyploidization that leads to larger genomes in ferns, but rather small-scale gene duplications (**Nakazato et al., 2006**). More recently, Marchant et al. published the first draft genome assembly for *C. richardii* and found additional support for genetic diploidy and limited rounds of polyploidization (**Marchant et al., 2019b**). These data further support the theory that homosporous fern genomes are large not because of whole-genome duplication, but because they do not have the same mechanisms for genome

downsizing as heterosporous plants (*Clark et al., 2016*; *Szövényi et al., 2021*).

The draft genome of *Ceratopteris richardii* is an important stepping stone for studying land plant evolution (*Marchant et al., 2019b*); a more complete genome is on the way that will be a better resource for genomic work (*Marchant et al., 2022*). A high-quality genome for *C. richardii* will also aid in the development of targeted enrichment or whole-genome resequencing. This latter advancement in sequencing resources for ferns will help us understand reticulate evolution and polyploidy in ferns, as phylogenies can be estimated with hundreds of genes.

In addition to the *Ceratopteris richardii* genome, there are many other fern genomes that have been recently published or will be available soon. Several heterosporous fern and lycophyte genomes have been published in the last few years, including the heterosporous ferns *Azolla filiculoides* and *Salvinia cucullata* (*Li et al., 2018*), and the heterosporous lycophytes *Selaginella moellendorffii* (*Banks et al., 2011*), *S. lepidophylla* (*VanBuren et al., 2018*), and *Isoëtes taiwanensis* (*Wickell et al., 2021*). In the near future, several additional homosporous fern genomes will be available, including *Adiantum capillus-veneris* (Polypodiales), *Alsophila spinulosa* (Cyatheales), *Dipteris conjugata* (Gleichinales), *Ptisana robusta* (Marattiales), *Huperzia asiatica* and *Diphasiastrum complanata* (Lycopodiales; Drs. F.-W. Li and M. Barker, personal communication). As more homosporous fern genomes become available, the preliminary work with the *C. richardii* genome will be tested in a more rigorous phylogenetic context, hopefully leading to a clearer picture of land plant genome evolution.

## Conclusion

Although *Ceratopteris richardii* has been used as a model for decades, fundamental aspects of its natural history are still unknown. A few examples include basic taxonomy, origins of spore number, salt tolerance, origins of polyploids, phenotypic plasticity, and intraspecies morphological variation (see *Box 2*). Many of these topics are ripe for undergraduate or graduate student projects and could be integrated into existing research programs to answer fundamental aspects of fern biology.

Additionally, *C. richardii* is a useful tool for teaching students at all grade levels about plant biology (https://www.c-fern.org/). As detailed by Marchant, the C-fern can be incorporated into curriculum topics ranging from basic plant

biology to evolution and development to bioinformatics. In the lab, field, or classroom, the recently published *C. richardii* genome provides a new window into the study of this fern (*Marchant et al., 2019b*). As more fern genomic resources become available, having *Ceratopteris* as a well-established model system will only become more important to test novel hypotheses about land plant evolution.

## Acknowledgements
We thank Jacob Suissa for the thoughtful comments on the manuscript. We are grateful to Christopher Haufler, David Randall, and Jo Ann Banks for providing the photograph of *Ceratopteris richardii* in cultivation.

**Sylvia P Kinosian** is in the Negaunee Institute for Plant Conservation Science, Chicago Botanic Garden, Glencoe, United States
sylvia.kinosian@gmail.com
http://orcid.org/0000-0002-0918-7196
 **Paul G Wolf** is in the Department of Biological Sciences, University of Alabama, Huntsville, United States
http://orcid.org/0000-0002-4317-6976

*Author contributions:* Sylvia P Kinosian, Conceptualization, Investigation, Visualization, Writing – original draft, Writing – review and editing; Paul G Wolf, Funding acquisition, Supervision, Writing – original draft, Writing – review and editing

*Competing interests:* The authors declare that no competing interests exist.

## Funding

| Funder | Grant reference number | Author |
|---|---|---|
| National Science Foundation | DEB-1911459 | Paul G Wolf |

The funders had no role in study design, data collection and interpretation, or the decision to submit the work for publication.

**Decision letter and Author response**
Decision letter https://doi.org/10.7554/eLife.75019.sa1
Author response https://doi.org/10.7554/eLife.75019.sa2

**Data availability**
Source data for Figure 2 (Range map of Ceratopteris) can be found in the file cer_locations.csv in https://github.com/sylviakinosian/ceratopteris-map (copy archived at swh:1:rev:02f4523dc32b20cb18b17e226eb6f2ff-b60cb05a) (previously published in Kinosian et al.,

2020a, https://doi.org/10.1016/j.ympev.2020.106938), and in Ceratopteris Brongn. in GBIF Secretariat (2021). GBIF Backbone Taxonomy. Checklist dataset https://doi.org/10.15468/39omei (accessed via GBIF.org on 2021-10-4).

The following previously published datasets were used:

| Author(s) | Year | Dataset URL | Database and Identifier |
|---|---|---|---|
| GBIF Secretariat | 2021 | https://doi.org/10.15468/39omei | GBIF, 10.15468/39omei |
| Kinosian SP, Pearse WD, Wolf PG | 2020 | https://github.com/sylviakinosian/ceratopteris-map | GitHub, 02f4523 |

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
