## [Decision Letter]

**Decision letter after peer review:**

Thank you for submitting your article "The Natural History of Model Organisms: The biology of the C-fern as a tool for understanding plant evolution" to *eLife* for consideration as a Feature Article. Your article has been reviewed by three peer reviewers, and the evaluation has been overseen by a member of the *eLife* Features Team (Helena Perez Valle). The following individuals involved in review of your submission have agreed to reveal their identity: Chi Lien Cheng and Verónica di Stilio.

The reviewers and editors have discussed the reviews and we have drafted this decision letter to help you prepare a revised submission.

Summary:

This short article on the fern model *Ceratopteris richardii* pulls together information about this fern's natural history, as well as reviewing its history as a model system. The use of understandable language and the framing of this species within the gametophyte-sporophyte relationship and the homospory-heterospory trends in plant evolution is welcome and refreshing. The article should be useful for those interested in learning more about ecological and genetic diversity in the genus, which could be leveraged in understanding genotype – phenotype relationships in this species.

Essential revisions:

1. Currently, the text is lacking in references on the breakthrough development of transgenic techniques in this model. Please discuss how the following references are a key aspect of this organism rising to model status:

– Plackett AR, Huang L, Sanders HL, Langdale JA. High-efficiency stable transformation of the model fern species *Ceratopteris richardii* via microparticle bombardment. Plant Physiol. 2014 May;165(1):3-14. doi: 10.1104/pp.113.231357. Epub 2014 Mar 12. PMID: 24623851; PMCID: PMC4012588.

– Plackett AR, Rabbinowitsch EH, Langdale JA. Protocol: genetic transformation of the fern *Ceratopteris richardii* through microparticle bombardment. Plant Methods. 2015 Jul 3;11:37. doi: 10.1186/s13007-015-0080-8. PMID: 26146510; PMCID: PMC4490597.

– Muthukumar B, Joyce BL, Elless MP, Stewart CN Jr. Stable transformation of ferns using spores as targets: Pteris vittata and *Ceratopteris thalictroides*. Plant Physiol. 2013 Oct;163(2):648-58. doi: 10.1104/pp.113.224675. Epub 2013 Aug 9. PMID: 23933990; PMCID: PMC3793046.

– Bui, L.T., Cordle, A.R., Irish, E.E. et al., Transient and stable transformation of *Ceratopteris richardii* gametophytes. BMC Res Notes 8, 214 (2015). https://doi.org/10.1186/s13104-015-1193-x

2. Please add a summary of the handful of functional studies that have been conducted since 2015 to illustrate the use of the transgenic techniques mentioned in the point above, mentioning how *C. richardii*'s natural history could be used to complement the knowledge that is emerging from these studies.

3. Please emphasize the potential use of *C. richardii* as a model to study plant evo-devo, based on ferns bridging the gap between the bryophyte systems (moss and Marchantia) and the angiosperms. For an example see the following review:

– Plackett AR, Di Stilio VS, Langdale JA. Ferns: the missing link in shoot evolution and development. Front Plant Sci. 2015 Nov 6;6:972. doi: 10.3389/fpls.2015.00972. PMID: 26594222; PMCID: PMC4635223.

4. Please update the photograph in Figure 1 to include a fern that is in a good state, preferably growing in the wild, the current plant depicted does not have the full set of leaves and does not look to be in a healthy state. If possible, please update the figure to also include photographs of several mutants. Please remember that any photographs used in the manuscript must be available to be published under a CC BY 4.0 license. Please contact me [h.perezvalle@elifesciences.org] if you have any queries about this.

5. If possible, please update Figure 2 to include photographs representing the diverse morphologies found in the different natural species, and update the figure caption to include details as to what questions each is most amenable for. Please remember that any photographs used in the manuscript must be available to be published under a CC BY 4.0 license.

6. Please include a figure with a simplified fern phylogeny to help a broader readership to comprehend the text better, with more details surrounding the genus *Ceratopteris* in the figure caption.

7. Please include further references regarding factors influencing sex determination, including light and hormones. In line 154, references from the Banks lab would be appropriate.

8. Please update the article to reflect that *C. richardii* is a semi-aquatic (not aquatic) species of fern, and if possible please discuss any work studying differences between aquatic and semi aquatic species of *Ceratopteris*.

9. The term "C-Fern" was trademarked by Leslie Hickock, so please use *Ceratopteris richardii* in the article (including the title), but please include a brief mention of the C-fern cultivar and its use in research.

---

## [Author Response]

Essential revisions:1. Currently, the text is lacking in references on the breakthrough development of transgenic techniques in this model. Please discuss how the following references are a key aspect of this organism rising to model status:– Plackett AR, Huang L, Sanders HL, Langdale JA. High-efficiency stable transformation of the model fern species *Ceratopteris richardii* via microparticle bombardment. Plant Physiol. 2014 May;165(1):3-14. doi: 10.1104/pp.113.231357. Epub 2014 Mar 12. PMID: 24623851; PMCID: PMC4012588.– Plackett AR, Rabbinowitsch EH, Langdale JA. Protocol: genetic transformation of the fern *Ceratopteris richardii* through microparticle bombardment. Plant Methods. 2015 Jul 3;11:37. doi: 10.1186/s13007-015-0080-8. PMID: 26146510; PMCID: PMC4490597.– Muthukumar B, Joyce BL, Elless MP, Stewart CN Jr. Stable transformation of ferns using spores as targets: Pteris vittata and *Ceratopteris thalictroides*. Plant Physiol. 2013 Oct;163(2):648-58. doi: 10.1104/pp.113.224675. Epub 2013 Aug 9. PMID: 23933990; PMCID: PMC3793046.– Bui, L.T., Cordle, A.R., Irish, E.E. et al., Transient and stable transformation of *Ceratopteris richardii* gametophytes. BMC Res Notes 8, 214 (2015). https://doi.org/10.1186/s13104-015-1193-x

Thank you for pointing out the lack of discussion on transgenic techniques in *C. richardii*.

We agree that this is an important part of this model, and have added a section titled, “*Ceratopteris* as a transgenic model for non-seed plants” that describes the various transformation approaches and how the life history of *Ceratopteris* enables the simple establishment of transgenic lines.

“Free-living generations and a flexible life cycle make *Ceratopteris* an important model for evolutionary developmental studies. […] The ability to have transgenic lines in both generations provides a unique perspective for studying how genes, growth conditions, or other factors affect sporophytes and gametophytes differently.”

2. Please add a summary of the handful of functional studies that have been conducted since 2015 to illustrate the use of the transgenic techniques mentioned in the point above, mentioning how *C. richardii*'s natural history could be used to complement the knowledge that is emerging from these studies.3. Please emphasize the potential use of *C. richardii* as a model to study plant evo-devo, based on ferns bridging the gap between the bryophyte systems (moss and Marchantia) and the angiosperms. For an example see the following review:– Plackett AR, Di Stilio VS, Langdale JA. Ferns: the missing link in shoot evolution and development. Front Plant Sci. 2015 Nov 6;6:972. doi: 10.3389/fpls.2015.00972. PMID: 26594222; PMCID: PMC4635223.

Thank you for these suggestions; we will address the two comments above here. We have added a discussion of transgenic techniques and how the natural history and phylogenetic placement of *Ceratopteris* makes it an important model for these studies:

“A recent study using both transgenic gametophytes and sporophytes of *Ceratopteris* investigated the role of the LEAFY transcription factor (LFY) in development (Plackett et al., 2018). LFY is important for cell division in moss embryos (Tanahashi et al., 2005) and angiosperm floral meristem development (Carpenter and Coen 1990). […] If a CRISPR system could be established in *Ceratopteris*, one might be able to extend such technology to other members of the Pteridaceae known for apogamy (Grusz et al., 2021; Grusz 2016; Grusz, Windham, and Pryer 2009), or for application in other non-model ferns.”

4. Please update the photograph in Figure 1 to include a fern that is in a good state, preferably growing in the wild, the current plant depicted does not have the full set of leaves and does not look to be in a healthy state. If possible, please update the figure to also include photographs of several mutants. Please remember that any photographs used in the manuscript must be available to be published under a CC BY 4.0 license. Please contact me [h.perezvalle@elifesciences.org] if you have any queries about this.5. If possible, please update Figure 2 to include photographs representing the diverse morphologies found in the different natural species, and update the figure caption to include details as to what questions each is most amenable for. Please remember that any photographs used in the manuscript must be available to be published under a CC BY 4.0 license.

Thank you for these suggestions, we will address the two comments above here. We have added a new figure (Figure 3) and included photographs of *C. thalictroides*, *C. pteridoides*, and *C. gaudichaudii*, in addition to a *C. richardii* sporophyte and gametophyte. We have included a new picture of *C. richardii* growing in cultivation; we are not aware of any recent (in the last 20 years) wild collections of *C. richardii* but would be very interested to see any.

6. Please include a figure with a simplified fern phylogeny to help a broader readership to comprehend the text better, with more details surrounding the genus *Ceratopteris* in the figure caption.

We have included a cartoon phylogeny (Figure 1) of land plants to place *Ceratopteris* in phylogenetic context. We have also updated Figure 2 to include a phylogenetic reconstruction of the current estimation of species relationships within the genus *Ceratopteris*. This was included in Figure 2 along with the distribution map of the genus to help the reader better contextualize the species range and relationships together.

7. Please include further references regarding factors influencing sex determination, including light and hormones. In line 154, references from the Banks lab would be appropriate.

Thank you for this suggestion, we have provided additional references throughout the manuscript where appropriate.

8. Please update the article to reflect that *C. richardii* is a semi-aquatic (not aquatic) species of fern, and if possible please discuss any work studying differences between aquatic and semi aquatic species of *Ceratopteris*.

We appreciate the reviewers pointing this out. We have made this distinction more clear throughout the manuscript. Little work has been done to investigate differences between the semi-aquatic and aquatic species of *Ceratopteris*. The two species most commonly found growing as fully aquatic plants are *C. pteridoides* (Americas) and *C. chingii* (endemic to China); the remaining species in the genus are mostly semi-aquatic, with the exception of *C. cornuta* which can sometimes be found growing unrooted in still or slow-moving water. The aquatic and semi-aquatic species can hybridize (e.g., *C. richardii* and *C. pteridoides*) and form morphologically intermediate offspring (Hickok and Klekowski, 1974). Additionally, a few gene expression studies have used *C. pteridoides*, but do not directly compare it to *C. richardii* or another semi-aquatic species. Researching potential differences between the semi-aquatic and aquatic species of *Ceratopteris* could be a great new avenue for work on fern ecology, gene function, or hybridization. We have added a short note to this extent in Box 2, on line 364:

“How does the habit (aquatic vs. semi-aquatic) of different *Ceratopteris* species influence population structure, breeding system, or genetic structure and function?”

9. The term "C-Fern" was trademarked by Leslie Hickock, so please use *Ceratopteris richardii* in the article (including the title), but please include a brief mention of the C-fern cultivar and its use in research.

Thank you for pointing this out. We have altered the title to be “The biology of *Ceratopteris richardii* as a tool for understanding plant evolution” and changed the usage of “C-fern” throughout the article to be *Ceratopteris richardii* or *C. richardii*. We have also added a mention of the C-fern cultivar on lines 58-9:

“Many studies used spores from a Cuban vouchered collection, now known as the Hnn strain or C-fern (Hickok 1977).”